# Systemic Minoxidil Accidental Exposure in a Paediatric Population: A Case Series Study of Cutaneous and Systemic Side Effects

**DOI:** 10.3390/jcm10184257

**Published:** 2021-09-20

**Authors:** Manuel Sánchez-Díaz, David López-Delgado, Trinidad Montero-Vílchez, Luis Salvador-Rodríguez, Alejandro Molina-Leyva, Jesús Tercedor-Sánchez, Salvador Arias-Santiago

**Affiliations:** 1Dermatology Unit, Hospital Universitario Virgen de las Nieves, IBS Granada, 18002 Granada, Spain; manolo.94.sanchez@gmail.com (M.S.-D.); davlopdel@gmail.com (D.L.-D.); tmonterov@gmail.com (T.M.-V.); l.salvador.rodriguez1991@gmail.com (L.S.-R.); alejandromolinaleyva@gmail.com (A.M.-L.); jesustercedor@gmail.com (J.T.-S.); 2Paediatric Dermatology Unit, Hospital Universitario Virgen de las Nieves, IBS Granada, 18002 Granada, Spain; 3Dermatology Department, School of Medicine, Granada University, 18002 Granada, Spain

**Keywords:** minoxidil, hypertrichosis, alopecia, children

## Abstract

Oral minoxidil is an approved treatment for high blood pressure which is also used as an off-label drug for alopecia. Knowledge about the effects of systemic minoxidil in the paediatric population is limited. A retrospective case series study of paediatric patients with history of systemic minoxidil intake due to contaminated sets of omeprazole was performed to describe side effects of high dose oral minoxidil intake in children. Twenty patients aged between 2 months and 13 years joined the study. They had received high doses of oral minoxidil (mean dose 0.90 mg/kg/day) during a mean time of 38.3 days. Hypertrichosis appeared in 65%, with a mean latency time of 24.31 days. Treatment time was associated with the appearance of hypertrichosis (*p* < 0.05). Most common initial zone of hypertrichosis was the face. Systemic effects developed in 15%, with no cases of severe disorders. The present study shows a novel insight into the side effects of high doses of oral minoxidil in children.

## 1. Introduction

Oral minoxidil is an approved treatment for high blood pressure. It is also used as an off-label drug for different types of alopecia [1,2,3]. Despite its generalized use, its mechanism of action for the treatment of alopecia remains unclear. Hypertrichosis, heart disorders and peripheral oedema are among its most frequent side effects. As this drug is mainly used in adult patients, the knowledge about the side effects of minoxidil in paediatric population is very limited.

In July 2019, a health alert was issued for the recall of several sets of omeprazole destined for master formulation, which contained minoxidil instead of omeprazole in Spain. The Spanish agency for drugs and medical products was the institution responsible for the health alert concerning sets intended for syrup or capsule formulation. This product was primarily intended for the paediatric population with gastroesophageal reflux, and was distributed in pharmacies in Cantabria, Andalucía and Valencia.

The aim of this study is to evaluate the side effects at skin and systemic levels of the exposure to systemic minoxidil in paediatric patients due to the intake of minoxidil from contaminated sets of omeprazole, and to explore associated factors related to the appearance of hypertrichosis and its latency time.

## 2. Materials and Methods

Design: A retrospective case series study was conducted.

Participants: Paediatric patients with history of systemic minoxidil intake due to contaminated sets of omeprazole treated at the Virgen de las Nieves University Hospital (Granada) was performed.

Data collection: When attending a protocolized medical revision, motivated by the accidental exposure to oral minoxidil, parents or legal representatives of the children were offered the option of participating in the study by answering a questionnaire which collected data related to the intake of minoxidil and its possible side effects. Data from complementary tests included in the medical protocol for the evaluation of these patients were also collected. All the parents or legal representatives of the patients gave their informed consent for participating in the study.

This study was approved by the Research Ethics Committee of the Virgen de las Nieves Hospital and is in accordance with the Declaration of Helsinki.

### 2.1. Inclusion and Exclusion Criteria

The inclusion criteria were as follow: (a) Paediatric patients aged between 0 and 14 years; (b) Informed consent from parents or legal representatives of the children with the agreement of the patient when it was possible; (c) Intake of any amount of omeprazole belonging to one of the contaminated sets. The belonging to one of the contaminated sets was verified through contacting the pharmacy where the drug was sold to check the batch code.

The exclusion criteria were as follow: (a) Patient’s refusal to participate in the study or the lack of informed consent from parents or legal representatives of the patients; (b) Patients who do not meet age criteria; (c) Patients with history of any disease which may cause hypertrichosis prior to the intake of minoxidil; (d) Patients who have taken any drug which may have as a side effect the appearance of hypertrichosis, before taking minoxidil or during the latency period until the appearance of hypertrichosis.

### 2.2. Variables of Interest

#### 2.2.1. Main Variables

Skin and systemic side effects of minoxidil intake were assessed by clinical examination, specific questionnaires concerning the onset and development of side effects, blood tests, electrocardiogram and echocardiography during the protocolized revisions in paediatric, dermatology and cardiology consultations:

Set of omeprazole which was administered, start date for the intake of omeprazole, duration of the treatment, dosage administered and total amount of minoxidil taken.

Skin side effects of minoxidil: The appearance of this side effects was assessed by the presence of hypertrichosis, latency time from the start of the treatment to the appearance of hypertrichosis, initial zone of appearance, areas affected, resolution of hypertrichosis when treatment was stopped, latency time from cessation of medication to resolution of hypertrichosis. Other possible skin effects: change in hair colour, Stevens–Johnson syndrome, dermatitis bullosa, skin rash, and toxic epidermal necrolysis.

Systemic side effects of minoxidil: heart disorders (tachycardia, pericarditis, pericardial effusion, cardiac tamponade, angina pectoris, electrocardiogram alterations); respiratory disorders (pleural effusion); hydrosaline retention, peripheral oedema or weight gain; gastrointestinal disorders; and any other alterations in complementary tests (liver and renal function tests, leukopenia, thrombocytopenia).

#### 2.2.2. Other Variables of Interest

Socio-demographic, biometric and clinical variables were recorded by clinical interview and physical examination (age, gender, weight and comorbidities).

### 2.3. Statistical Analysis

Descriptive statistics were used to evaluate the characteristics of the sample. The Shapiro–Wilk test was used to assess the normality of the variables. Continuous variables are expressed as mean and standard deviation (SD). Qualitative variables are expressed as relative and absolute frequency distributions.

The χ2 test or Fisher’s exact test, as appropriate, were used to compare nominal variables, and the Student’s t-test or Wilcoxon–Mann–Whitney test were used to compare between nominal and continuous data. Statistical significance was considered if *p* values were less than 0.05. Statistical analyses were performed using JMP version 9.0.1 (SAS institute Inc., Cary, North Carolina, USA).

## 3. Results

### 3.1. Sociodemographic and Clinical Features of the Sample

Twenty-one patients were invited to join the study, 95% (20/21) of them gave their informed consent and agreed to participate, so twenty patients were included. Mean age was 3.89 years old, and male to female ratio was 1.5:1. None of the patients had medical history of any diseases related to hypertrichosis prior to minoxidil intake. The indication for omeprazole treatment was mostly gastroesophageal reflux: 75% (15/20). Other indications were Helicobacter Pylori eradication treatment: 15% (3/20); and gastric prophylaxis during steroid treatment for nephrotic syndrome and Kawasaki disease: 10% (2/20). Summary of clinical characteristics can be seen in Table 1; complete data are shown in Table A1 and Table A2.

Hypertrichosis appeared in 65% (13/20) of the patients with a wide range of latency time from the start of the treatment. The initial zone of appearance was the face, mostly forehead and temples 61% (8/13). Other first locations were the back and the limbs. During evolution, hypertrichosis finally developed in the facial area in all patients. In the follow-up, all patients with hypertrichosis developed generalized hypertrichosis as well (Figure 1). Resolution within 6 months of follow-up was observed in 61.5% (8/13) of the patients, with an 38.5% (5/13) of the patients showing persistent hypertrichosis after this time.

Other skin side effects appeared in 10% (2/20) of the patients: changes in the hypertrichotic hair colour, appearing lighter hair in one case and darker hair in another. It developed only in the areas affected by the hypertrichosis, with no changes in the colour of the rest of the hair. No cases of Stevens-Johnson Syndrome nor toxic epidermal necrolysis appeared.

Regarding systemic side effects, no severe disorders were found. Only 15% of the patients (3/20) developed systemic side effects: One of them presented with diarrhoea and anxiety, another with headache and facial oedema and the last one with severe asthenia. These manifestations appeared during the treatment, and were transient, resolving within three days. Their relationship with the intake of minoxidil could not be assured.

### 3.2. Factors Associated to the Appearance of Hypertrichosis

Median values of doses and treatment times in hypertrichosis and no hypertrichosis patients, as well as statistic tests can be seen in Table 2. No differences were found in daily dose, accumulated total dose or adjusted daily dose between patients with and without hypertrichosis. Moreover, hypertrichosis was not associated with sex or age. However, treatment duration was found to be associated with the appearance of hypertrichosis (*p* = 0.028), showing that patients with hypertrichosis had had longer treatments.

## 4. Discussion

The main side effect observed in our series was the hypertrichosis, with a low rate of systemic side effects despite high doses of minoxidil maintained over a long period of time. The long period of time, as well, was significantly different between both groups, finding longer treatments associated to hypertrichosis. These data provide new information on the biological behaviour in the paediatric population of a well-established treatment for alopecia as minoxidil. The knowledge about the effects of minoxidil on children might be helpful to improve the use of minoxidil in paediatric patients with severe alopecic disorders, such as alopecia areata [4].

Minoxidil was first introduced as an oral treatment for severe hypertension in the 1970s [5]. When hypertrichosis was observed as a side effect of this medication, both topical and oral formulations of minoxidil were developed to treat alopecia. Despite this effect has been known for over 30 years, the mechanism of action through which minoxidil produces hair growth is not yet fully understood.

This drug acts as an arteriolar vasodilator which opens potassium channels of the smooth muscles of peripheral arteries. This leads to hyperpolarization of the cells, causing vasodilation and decreasing blood pressure. Regarding the hair follicle, minoxidil is thought to shorten the telogen phase and to extend the anagen phase [6]. In histological studies, an increase in anagen follicle percentage and bigger follicles size was noted [7]. At the biochemical level, minoxidil is able to induce several changes in the expression of different molecules: Minoxidil increases vascular endothelial growth factor (VEGF) level, hypoxia-inducible factor-1-alpha, and prostaglandin E2 production [8,9,10]. These biochemical changes may modify the micro-environment of the hair follicle, inducing a prolonged anagen phase and increased hair growth (Figure 2). It is thought that the action of minoxidil on hair growth is mainly due to minoxidil sulphate, a metabolite of minoxidil. The enzyme responsible for its generation is sulfotransferase, which is located in hair follicles [6]. There is a variety in production of sulfotransferase among individual, which could explain the differences in the response after minoxidil treatment observed among patients, with patients having higher enzyme activity showing better response to minoxidil [6]. Moreover, there is no correlation between concentration of minoxidil in blood or tissues and hair growth [11]. As salicylate and aspirin can inhibit sulfotransferase, prior or concomitant intake of these drugs may lead to a decreased clinical response to minoxidil [12].

Most of the reports focus on the effects of topical minoxidil in children [13] which is generally a well-tolerated treatment. However, there are reports of systemic toxicity in children due to percutaneous absorption of topical minoxidil [14]. Regarding oral minoxidil and children the current scientific evidence is limited to a report of the accidental ingestion of minoxidil [14], a randomized clinical trial with 9 patients receiving oral minoxidil treatment for arterial wall hypertrophy for Williams–Beuren syndrome [15], and some case series about the use of oral minoxidil for refractory hypertension in children [16,17,18]. In addition, a recently published systematic review [4] addressed the issue of minoxidil and children, finding only reports of topical minoxidil in children and one report of low-dose oral minoxidil in a 11-years-old female. To the best of our knowledge, apart from a recent study, which addressed the treatment with low-dose oral minoxidil in children with loose anagen hair syndrome [19], there is a lack of evidence of the use of oral minoxidil in children.

The mentioned studies agree on the possible side effects of minoxidil in paediatric population, hypertrichosis is the most common, and fluid retention, arrhythmia, electrocardiogram alterations and pericarditis are some of the most severe [13,14,15,16,17,18].

The initial dose of oral minoxidil in paediatric population regarding the treatment of hypertension is 0.1–0.2 mg/kg/day. It may be increase up to 1 mg/kg/day depending on the tolerance and effectiveness of the treatment [15,20]. When used to treat loose anagen hair syndrome [19], dose administered was lower than 0.02mg/kg/day. Severe adverse events seem to be dose-dependent, with a positive gradient [13,14,15,16,17,18]. However, most of these studies have a small sample size and none of them focus on the onset, latency, location and evolution of hypertrichosis.

Doses taken by the patients included in our study were relatively high in comparison to the maximum doses used when treating hypertension and loose anagen hair syndrome in children. The highest dose was taken by the patient number 1, more than 2 mg/kg/day. The dosage schedule ranged between 0.33–2.12 mg/kg/day.

No differences were found between hypertrichosis and no hypertrichosis groups in terms of daily dose, daily adjusted dose and accumulated dose in our study. Individual variations in sulfotransferase might explain the fact that the dose of minoxidil does not seem to be related to the development of hypertrichosis. In accordance with this hypothesis, patients with increased sulfotransferase activity would develop hypertrichosis after taking minoxidil for a sufficient period of time, regardless of the dose administered.

The low rate of systemic effects in our study seem to differ from other reports [6,7,8,9,10]. This fact may correspond to differences in the posology and dose of minoxidil administered. The short duration of the treatment in several cases might have led to the appearance of less systemic side effects.

A recent study [21] addressed the issue of low-dose minoxidil-induced hypertrichosis in adult patients, showing a mean time between the start of the treatment and the hypertrichosis of 2 months, and also a high percentage of patients with facial hypertrichosis. On the one hand, the location of the hypertrichosis is consistent with the findings in our study. On the other hand, the shorter latency time in our study may be due to the higher doses of minoxidil administered and the younger age of the patients. High doses of minoxidil might also be responsible for the long resolution time of hypertrichosis in our case series.

The main limitations of this study are its small sample, its retrospective nature, and the possible selection bias committed by collecting only those patients who consulted because of the accidental minoxidil intake.

## 5. Conclusions

We report high-dose minoxidil adverse events in a group of paediatric patients. The facial area as the first affected location, the high proportion of patients showing long-lasting hypertrichosis are facts which show a novel insight into the effects of oral minoxidil in the paediatric population. Moreover, treatment time seem to be related to the appearance of hypertrichosis. This information may be useful for further studies and clinical applications involving the use of topical or systemic minoxidil in children with alopecia.

## Figures and Tables

**Figure 1 jcm-10-04257-f001:**
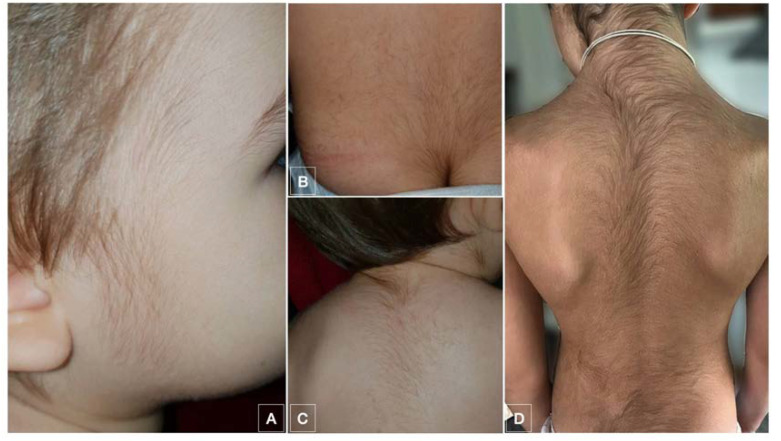
Images showing hypertrichosis which appeared in patient 9 (**A**–**C**) and 15 (**D**).

**Figure 2 jcm-10-04257-f002:**
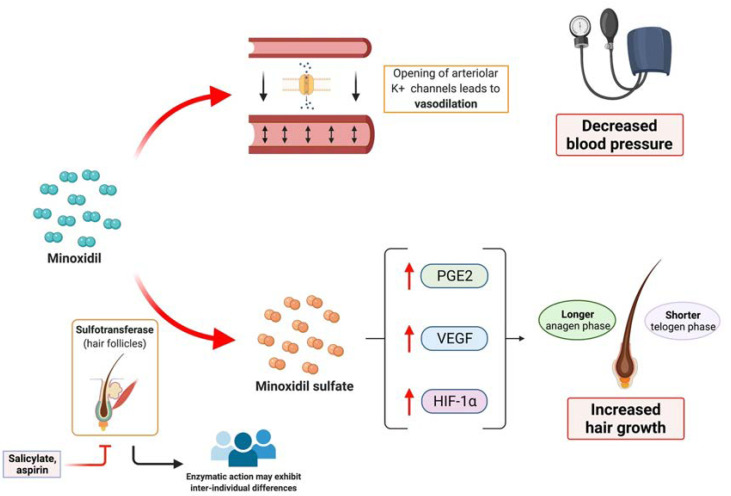
Overview of the mechanism of action of minoxidil. Minoxidil induces the opening of arteriolar potassium channels, leading to vasodilation, which is the basis of its utility regarding the treatment of hypertension. Minoxidil also induces biochemical changes within the hair follicle, such as the increase of vascular endothelial growth factor (VEGF), hypoxia-induced factor 1 alfa (HIG-1alfa) and prostaglandin E2. These changes lead to a longer anagen phase and a shorter telogen phase, inducing an increased hair growth. Sulfotransferase, which can be inhibited by salicylates and aspirin, is responsible for generating minoxidil sulphate, which is the active form of minoxidil within the hair follicles. Inter-individual differences in sulfotransferase could explain the different responses to treatment. Created with BioRender.

**Table 1 jcm-10-04257-t001:** Overview of the characteristics of the sample.

Variable	Mean (SD)/% (*n*/*N*)
*N* = 20 patients	
Sex	Male: 60% (12/20)
	Female: 40% (8/20)
Age (years)	3.89 (SD 3.82)
Weight (kg)	16.29 (SD9.71)
Duration of treatment (days)	38.3 (SD 38.68)
Daily dose (mg/24 h)	13.22 (SD 8.23)
Adjusted dose (mg/kg/day)	0.90 (SD 0.43)
Accumulated dose (mg)	453.75 (SD 513.36)
Appearance of hypertrichosis	Yes: 65% (13/20)
	No: 35% (7/20)
Onset latency time for hypertrichosis (days)	24.31 (SD 19.77)
Resolution of hypertrichosis	Yes: 61.5% (8/13)
	No: 38.5% (5/13)
Onset latency time for resolution (days)	76.25 (SD 29.25)

“*N*”: Total sample; “*n*”: number of patients in each category.

**Table 2 jcm-10-04257-t002:** Median values of the variables. Significant statistical differences were found in the median treatment time between hypertrichotic and no hypertrichotic patients.

Patient Group	Hypertrichosis	No Hypertrichosis	*p* Value
Median total daily dose (mg/day)	10	10	0.497
Median adjusted daily dose (mg/kg/day)	0.85	0.91	0.395
Median accumulated total dose (mg)	300	240	0.13
Median treatment time (days)	30	12	0.028

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
