# Peer review of "Systemic Minoxidil Accidental Exposure in a Paediatric Population: A Case Series Study of Cutaneous and Systemic Side Effects"

_jcm, 2021, doi:10.3390/jcm10184257_

Round 1

Reviewer 1 Report

This is a very interesting paper regarding the side effects of the use of oral minoxidil in children. As minoxidil is becoming an option for the treatment of different types of alopecia, a better understanding of its side effects is advisable, especially in paediatric population, where the available information is limited. Moreover, the fact that the dose taken by the patients was higher than the dose used in alopecia and despite that the patients did not have any severe systemic disorder, may provide dermatologists with more confidence about its security for its use in alopecia.

I would suggest some minor changes to the authors:

- The presence of persistent hypertrichosis is rather interesting, so most adults who receive minoxidil and develop hypertrichosis usually have resolution of the hypertrichosis when reducing the dose of minoxidil or stopping the treatment. Do the authors have any comment about the possible reason of that persistence?

- There are some minor English language mistakes.

Author Response

Dear Reviewer, 

We would like to thank you for your comments, as they allow us to improve our scientific work. All the commentaries made have been taken into account. Below there is a point-by-point response to your comments: 

1) The real cause of the late resolution of the hypertrichosis is an issue to be addressed in further studies. However, apart from possible differences between children and adults regarding hair follicle metabolism, the high doses which have been administered to the patients in our case series could be the key to understand the long duration of hypertrichosis. A sentence has been added in the text to address this issue. 

2) English has been corrected. 

Reviewer 2 Report

A very interesting and disturbing story, and at the same time a "medical experiment".

It should be clearly stated in the work whether:

1 / false omeprazole batches were verified(?)

2 / what institution detected monixidil,

3 / whether minoxidil was measured in the blood of children(?)

4 / are omeprazole tablets or caps. administered from 2 months of age?

And also:

5 / Line 177:  in the text – „topical minoxidil” but Ref. 13 - only about systemic? Please change the ref.

6 / Line 224: "a large" but only !!! N = 20.

7 / Some problems in Ref. 19 and 21 and in some others

Author Response

Dear Reviewer, 

We thank you very much for your comments, as they allow us to improve our work. We have changed all the required statements. Here, there is a point-by-point response to your comments: 

1) As it has been added to the text, "false" sets of omeprazole were verified.

2) The institution which detected minoxidil was the "AEMPS" ("Agencia Española de Medicamentos y Productos Sanitarios" - Spanish Agency for drugs and sanitary products). It has been added to the text. 

3) Minoxidil was not detected in blood as that test is not available in our Hospital. 

4) Omeprazole was made by magistral formulation as a syrup, not tablets or caps. It has been added to the text.

5) References has been corrected. 

6) The sentence has been rewritten. 

7) References has been corrected.